# Self-Reflection and Peer-Assessments Effect on Pharmacy Students’ Performance at Simulated Counselling Sessions

**DOI:** 10.3390/pharmacy11010005

**Published:** 2022-12-27

**Authors:** Andrew Bartlett, Jessica Pace, Angela Arora, Jonathan Penm

**Affiliations:** 1School of Pharmacy, Faculty of Medicine and Health, The University of Sydney, Camperdown, NSW 2006, Australia; 2Department of Pharmacy, Prince of Wales Hospital, Randwick, NSW 2031, Australia

**Keywords:** patient counselling, feedback, self-reflection, peer assessment, self-assessment

## Abstract

Introduction: Verbal communication is a vital skill for pharmacists and essential for improving patient care. The aim of this study was to explore students’ perception of the impact of self-reflection and self- and peer-assessment on simulated patient counselling sessions. Methods: Focus groups explored student perceptions of how this course and way of learning has impacted their performance at counselling patients. Data were analysed using iterative inductive thematic analysis procedures and mapped to the self-determination theory. Results: Nine focus groups with 47 pharmacy students. We identified three main themes and ten associated subthemes. These were learning style (sub-themes gradual introduction to assessment, learning through self-reflection videos, authentic assessment, individual learning compared to group learning, and learning through observation of best practice), feedback (sub-themes inconsistent feedback, summative feedback, perception of self and relationship with peers informing peer assessment) and benefits in real life practice. These themes mapped well to self-determination theory and highlighted that additional focus may be required for benefits in real-life practice. Conclusion: Students’ perceptions of self-reflection and self- and peer-assessment centred on learning style, feedback, and benefits in real-life practice. Additional focus on benefits of this unit of study in real-life practice and work integrated learning on placements may further strengthen the impact of these learning activities.

## 1. Introduction

Proficiency in verbal communication is a vital skill for pharmacists and is essential to improve patient care. For example, the World Health Organisation (WHO) [1] emphasises the importance of pharmacists as communicators while the International Pharmaceutical Federation (FIP) competency standards [2] emphasise a range of skills related to verbal communication with patients, including counselling the patient on the safe and rational use of medicines and devices; communicating effectively with patients, using lay terms and checking understanding; tailoring communication that is appropriate to the patient’s needs; and using appropriate communication skills to establish and maintain rapport with the patient when communicating through digital and electronic platforms. Meanwhile, in Australia (the setting of the educational intervention examined in this paper), a range of professional standards, codes and guidelines highlight the importance of communication skills for both students and practising pharmacists [3,4,5,6].

Simulation is one approach that that is widely used in pharmacy education for teaching patient counselling skills. A range of simulation methods have been used to teach patient counselling skills, including role-play with simulated patients or actors [7,8,9,10,11], peers [12,13,14,15,16], and faculty members [17,18,19,20], video recording of simulated communication [15,16,21,22], high-fidelity simulation [23,24], and mystery shoppers [25]. However, the most appropriate method, how this is used and the mechanism by which it achieves its effects depends on the stage of the student’s learning trajectory [26]. For example, it has been proposed that role play with simulated patients works for pharmacy students at all levels through practice and consistent feedback to improve communication and self-awareness. Meanwhile role play with peers works for intermediate and senior students through repeated practice, repetition and consistent feedback to increase communication confidence through reflection and self-assessment. However, this may not work for junior students due to peer feedback causing discomfort and limiting reflection.

It has been suggested that student-centred strategies such as self-reflection and peer assessment may further enhance students learning and counselling skill development [27]. Self-reflective sessions especially among peers in a classroom setting provides learners opportunities to develop the soft skills of self-identification of gaps which must be filled to make progress. The use of peer assessment and self-reflection sessions may also complement each other to quicken this process and allow the learner to identify areas of need that they may not have been able to identify on their own [28]. There is a growing literature on the use of self-reflection and peer assessments in pharmacy education. For example, recent work examined the accuracy of pharmacy students’ self-assessment skills [29] while others looked at ways to assess self-reflection activities [30] as well as initiatives that incorporate self-reflection activities—including those relating to pharmacy—into pharmacy courses [31] Meanwhile, the literature on peer assessment primarily consists of a description and evaluation of its use in particular learning settings and for specified learning activities [32,33] There is little evaluation of the effect of the combination of self-reflection and peer assessment in simulated patient counselling activities in pharmacy.

As teaching is a moral and educational enterprise, teachers should be expected to explain and justify the instructional methods and designs they use in their teaching [34]. Therefore, the aim of this study is to explore students’ perception of the impact of self-reflection and self- and peer-assessment on simulated patient counselling sessions.

## 2. Materials and Methods

This study included focus groups with pharmacy students about the use of self-reflection and peer assessment at simulated counselling sessions on their learning experiences. Ethics approval for this project was received from the University of Sydney’s Human Research Ethics Committee (protocol number 2018/603).

### 2.1. Study Setting

All third-year students in the University of Sydney Bachelor of Pharmacy degree are required to enrol in PHAR3825 Pharmaceutical Skills and Dispensing B. This unit of study consolidates previous units in the curriculum, building upon their therapeutic knowledge to counsel patients on the appropriate use of prescribed medications. In first- and second-year students have participated in simulated patient counselling interactions with their peers on Over-the-Counter medicines. In 2018, the unit was designed so that each student individually completed seven simulated patient counselling scenarios with a demonstrator. With their peers, they also both completed and observed their peer completing 28 simulated patient counselling scenarios (from which the seven cases completed with a demonstrator are drawn); in the peer exercises, the student who was not completing the counselling scenario marked their peer using a provided grading rubric. Students could choose which peer they completed and observed the case studies with (i.e., peer evaluations were identified and not assigned) and there was no screening of comments before release.

Case studies were drawn from the top 50 Pharmaceutical Benefits Scheme (PBS) dispensed drugs by volume [35] and students were provided with a standard structure to use when counselling patients, whereby counselling is based around “three prime questions” [36]—(1) “what did the doctor tell you this medicine is for?”, (2) “how did the doctor tell you to take the medicine?” and (3) “what did the doctor tell you to expect?”. Each session, students were also required to make a video of themselves counselling a peer on one of the cases and then mark this using a standard grading rubric and to self-reflect on their cases to identify their own unique learning needs; self-reflection occurs both by completing a standard set of reflection questions after each counselling session and by completing a reflective statement based around a standard set of reflection questions at the end of the unit (see Figure 1). The learning outcomes primarily included effective counselling and education of patients about medicines and disease states.

Each year, there are around 240 third year Bachelor of Pharmacy students enrolled in this unit of study, with the majority being in their early 20s. All enrolled students were invited to participate in the focus groups via announcements in class and the Canvas Learning Management System used in this unit; those interested were invited to contact the research team and a suitable time for focus groups was arranged. There were no dropouts and it was envisaged that up to 30 students would participate. As per previous studies [37], this is a well-accepted sample size in qualitative studies and is sufficient to achieve thematic saturation and therefore adequately explore variation in experiences amongst participants.

### 2.2. Data Collection

Nine focus groups with 47 students (15 male and 32 female, range one to eight students in each focus group) were conducted by JP, JB, SC, SH and SK in October 2018 and October 2019. Three interviewers were male and two were female. All were experienced pharmacy academics with qualitative research training and experience running focus groups that were not teaching into this unit of study. However, students may have had prior relationships with these academics from other units in the degree. Focus groups lasted for an average of 45 min (range 35 to 55 min) and were conducted at the University of Sydney campus. Only the participants and researchers were present for focus groups. Focus groups explored student perceptions of how this course and way of learning has impacted on their individual learning outcomes and preparedness to be a pharmacist. All focus group interviews were audio recorded (with the participants’ permission) and transcribed verbatim by AA; field notes were also taken. See Appendix A for focus group interview guide. The interview guide was developed after discussion by the academic team as an evaluation of the course and did not use a specific framework.

### 2.3. Data Analysis

A two-phase pragmatic approach was taken to data analysis. Phase one involved thematic analysis using an inductive approach. Data were checked for accuracy and entered into QSR International Nvivo software version 12 for data management and analysis. Each participant was assigned a number for anonymity. A phenomenological approach underpinned the data analysis. This seeks to describe the essence of a phenomenon by exploring it from the perspective of those who have experienced it, with the aim of describing the meaning of the experience, both in terms of what was experienced and how it was experienced [38].

Data were analysed using iterative inductive thematic analysis procedures as outlined by Braun and Clarke [39]. This method of analysis is used for identifying, analysing and reporting patterns from data sources and developing interpretations of those patterns. Initial data coding and analysis were conducted by AA along with discussion amongst the authors to refine codes during this interpretive process [40]. This allowed the authors to observe significant patterns in the data, enabling an understanding of commonalities in participants’ responses. The analysis process involved six steps. First, all members of the research team familiarized themselves with data, e.g., by listening to recordings and reading transcripts. Initial codes were then generated by AA and reviewed by AB and JP. Ongoing discussion amongst JP, AB and AA was used to refine the codes and develop, define and name themes. Data analysis continued until all transcripts were analysed and thematic saturation [41] (the point at which no new themes are emerging and all themes are complete and well-described) was reached. JLP produced the final report which was reviewed and approved by all authors.

A second phase was undertaken whereby the inductively derived themes were mapped to Ryan and Deci’s self-determination theory [42]. This theory suggests that people are motivated to grow and develop by three innate universal psychological needs: autonomy (people need to feel in control of their own behaviours and goals and the sense that being able to take direct action will result in real change plays a major part in helping people feel self-determined); competence (people need to gain mastery of tasks and learn different skills and when they feel that they have the skills needed for success, they are more likely to take actions that will help them achieve their goals); and relatedness (people need to experience a sense of belonging and attachment to other people). Here, we aimed to use mapping of themes against this framework to identify areas for further improvement in the use of self-reflection and peer assessments to improve patient counselling by pharmacy students.

Although participants had the opportunity to review findings, none requested this and so no feedback was given.

We followed the Consolidated Criteria for Reporting Qualitative Research (COREQ) checklist [43] when reporting this study.

## 3. Results

Identified themes as mapped against the components of self-determination theory are outlined in Table 1.

We identified three main themes and ten associated subthemes. These are learning style (sub-themes gradual introduction to assessment, learning through self-reflection videos, learning through authentic assessment, individual learning compared to group learning, and learning through observation of best practice), feedback (sub-themes inconsistent feedback, summative feedback, perception of self and relationship with peers informing peer assessment) and benefits in real life practice. We describe each of these themes in more detail and outline how they map to self-determination theory below.

### 3.1. Learning Style

Students identified two key factors related to learning style that impacted their autonomy—gradual introduction to assessment and learning through self-reflection videos.

#### 3.1.1. Gradual Introduction to Assessment

Firstly, students noted that they appreciated the gradual introduction to assessment in the unit, where assessment is formative in the early sessions for all components and then summative assessment is gradually introduced (here, there is summative assessment for labels for dispensed products from session 2 onwards and for counselling from session 3 onwards). In this way, students could adjust to the expectations and mode of assessment in the unit before undertaking summative assessments later in the course.


*That was nice how kind of gradually ramped up like it started off as it wasn’t assessed and then the next [one] we did, just the labels and then you did the labels and the counselling and then you do counselling for all of them.*
Student 5, focus group 6.

#### 3.1.2. Learning through Self-Reflection Videos

Students also outlined several ways that self-reflection on videos of their counselling sessions helped them to develop their skills in patient counselling. Firstly, they noted that when they are counselling, they cannot observe themselves or their non-verbal communication. The video allows them to go back and look at how they performed and where they need to improve.


*I find the video, like the one video is actually quite helpful because you can go back and look at your expressions and your body language and that sounds.*
Student 3, focus group 7.

Additionally, many noted that it provided a more realistic perspective on their counselling. For example, they noted that when observing their peers, they did not understand why their peers approached the counselling in a particular way or made certain mistakes. However, after watching their own videos back, they realised that not everyone is perfect, and everyone makes mistakes in their counselling.


*That thing where you see someone else or you’re like Oh but why’d they do this. But then when you see a video of yourself you like right there and that’s where you like you know you do make mistakes.*
Student 3, focus group 9.

Finally, students appreciated that the videos gave them a chance to practice counselling for a particular product in a non-judgmental environment, as they could just restart the tape if they made a mistake (and this was less stressful than practising live with a peer or tutor). Saying that, they expressed that this could become tedious, particularly if they were required to mark each attempt using the grading rubric.


*In saying that, the video was good because during the video you would stuff up, you redo it, redo it chance to practice other than the actual scenarios themselves whichever so this is a good way to practice it without having the pressure of the marks there but then having to get the grading criteria that’s a bit tedious.*
Student 5, focus group 8.

They also identified two factors related to learning style that affected their competence—the use of authentic assessment and individual compared to group learning.

#### 3.1.3. Learning through Authentic Assessment

Students expressed a desire for authentic assessments and there were mixed views on whether the counselling assessments achieved this. Some students expressed that these mirrored the patient counselling that they will complete in practice well, noting that when talking to a patient they would not always have access to key pharmacy texts, and it was valuable to practice in class without these.


*Not having every resource there it’s just more of what you find in class and what you remember and forcing yourself to do that. Is going to help you in a practical sense where you don’t always have your AMH on you when you’re talking to a patient. And so this is a very realistic exposure to that sort of situation and did help from the first week.*
Student 3, focus group 7.

Others questioned the value of these assessments, expressing that the focus was on memorisation of either a general scaffold or relevant ancillary labels instead of a quality counselling interaction that would be useful to the patient. They instead expressed a desire for more problem-based learning, where they would be given a problem scenario and would need to use their resources to solve a clinical problem before conveying their findings and advice to the patient. They also felt that this would better help them to prepare for another major assessment that they complete in this year of their degree—the objective structured clinical examination (OSCE).


*If our week-to-week scenario was more like an OSCE question where we would be given the scenario, we are able to browse through our resources and find the answer instead of just having to memories all the ancillary labels.*
Student 5, focus group 8.

#### 3.1.4. Individual Learning Compared to Group Learning

Students also expressed a desire for individual-based rather than group-based learning to improve their counselling skills. They felt that this forced students to take responsibility for their own learning, for example by completing prework to adequately prepare for class and practising each scenario. This is in comparison to group-based learning, where it is easier for students to hide behind other students who complete required tasks, or they need to rely on other students to complete the required work.


*I feel like a lot more people were responsible in doing the prework and coming prepared, I found a lot more people were prepared then if it was group-based learning where it would be easier to rely on another person.*
Student 1, focus group 1.


*I liked that it actually helps you, like it forces you to practice…which is really beneficial for a lot of people.*
Student 2, focus group 6.

#### 3.1.5. Learning through Observation of Best Practice

Finally, students expressed a desire to learn through observation of best practice to increase their sense of relatedness (both to other more experienced pharmacists and the overall pharmacy profession). This would both show them the standard expected and give them ideas on how to approach their own counselling. They acknowledged that they had limited time in their counselling classes and that this should primarily be used for them to practice their counselling skills but suggested either that demonstrators could start each session by demonstrating a counselling session or a video of a demonstrator counselling could be uploaded to the online learning management system for them to view as part of their prework before coming to class.


*I like to listen to someone do it first and I like to be like Oh I like that and then I’ll take it. But I don’t know, I feel like maybe a suggestion like I know we don’t have that much time during labs but maybe if two demos just demonstrate what is a good counselling session because I’ve never really heard what’s good like what is good.*
Student 6, focus group 6.

### 3.2. Feedback

#### 3.2.1. Inconsistent Feedback

Inconsistent feedback between demonstrators was the main way that feedback affected their autonomy. They noted inconsistencies in the areas that demonstrators focused on when marking, the perceived standard to which demonstrators marked (with some demonstrators seen to be “stingier” than others) and the level of prompting that occurs during summative assessments. These perceived inconsistencies left students feeling frustrated and not in control of their learning in the unit.


*Some tutors prompt you and some don’t and then that determines whether you pass or fail.*
Student 5, focus group 4.


*So, it’s a bit of discrepancy between what assessors are looking for as well which I found a bit frustrating to kind of deal with as well.*
Student 2, focus group 6.

#### 3.2.2. Summative Feedback

Students identified summative feedback as important at improving their competence in this unit. They noted that being marked on the counselling scenarios rather than just practising as an in-class activity that does not count towards their final marks as occurs in other units was very useful. It meant that students both took the activities more seriously than they otherwise would have and took more time to prepare for these.


*I thought that was also really beneficial in comparison with the other ones, actually having marked, being marked on umm on our performance was also very useful that meant we took it more seriously, we took more time to prepare for the, for the medications like, for the scenarios before we came in.*
Student 1, focus group 1.

While students noted that summative feedback did not detract from their enjoyment of the unit, they did acknowledge that it added a level of stress, for example through the knowledge that their performance counted towards their final grades and the time pressure imposed during the counselling assessments.


*I love it, it’s actually really fun like one of my fun unit out of all the units that we have been doing, because you kind of interact, like you get to do something rather than sitting down and writing with pens and what not but it’s kind of stressful at the same time because you are under time pressure.*
Student 3, focus group 4.

#### 3.2.3. Perception of Self and Relationship with Peers Informing Assessment

Finally, students identified two ways that feedback relates to relatedness. Importantly, here they focused on the impact of their view of themselves and relationships their peers on feedback provided, rather than on their relationships more experienced colleagues and the broader pharmacy profession as we found for learning style. Students noted that self-assessment impacted their perception of self and their own performance. Viewing their videos allowed them to immediately identify gaps in their knowledge while marking their performance against the grading rubric further helped them to evaluate their own performance.


*Watching my video I do immediately see and see gaps in knowledge or like if I’m comparing if I’m watching it while looking at the rubric I can see things that I don’t.*
Student 2, focus group 9.

However, they also noted the subjective nature of peer assessment and their tendency to mark themselves or a stranger more harshly than they would someone they considered a friend.


*But I mean the self-assessment also is like really subjective like I am such a pessimist and I’m really hard on myself. Whereas if I watch, say Sally and I did the exact same thing then I would probably give, I would give her a higher mark than what I would give myself even if we said the exact same thing.*
Student 4, focus group 6.

Many noted that they did not take giving feedback to peers as seriously as they did when critiquing themselves. As a result, feedback to friends was often not as honest, comprehensive or critical as it could be.


*So if I had a friend, I found that it would be very casual, their feedback wouldn’t be umm as honest or comprehensive in comparison when it was with a random person who took the scenario more seriously and who wanted to give feedback.*
Student 1, focus group 1.

As a result, some students questioned the usefulness of peer feedback and instead preferred watching their own videos and marking these against the grading rubric as the main feedback used to evaluate their counselling performance.


*I’d prefer the video. The video alone that would be fantastic but the peer marking me that’s just we are all mates here, we would give each other 100%. I would give anyone 100%, I wouldn’t take that seriously though.*
Student 5, focus group 8.

#### 3.2.4. Benefits in Real Life Practice

Students identified a number of benefits of the counselling assessments to their real-life practice; all of these related to their competence at counselling patients in everyday pharmacy practice. Firstly, students appreciated the opportunity to practice and develop skills important for patient counselling—such as the ability to think on their feet and adapt their counselling style and content to different patients—in a realistic practice environment.


*I think it’s more effective, because it’s putting us in what is similar to reality situations and it kinda puts us in the spotlight, we kinda have to do, we do have to think on our feet, it is a trait we do need to learn for the future.*
Student 3, focus group 3.

Students noted that the counselling sessions both increased their knowledge of the key counselling points for commonly dispensed and helped them to consolidate the knowledge that they have gained from other units of study in their pharmacy degree and apply this to patient counselling. They found both of these useful when counselling patients in their everyday pharmacy practice.


*Even at like work I remember all the counselling points from what I learnt here, and I can say to people in real life. And it really helps them too, like I like how you guys like teach us like the main points of it is like I think it really helps.*
Student 5, focus group 7.


*For consultations I actually found it more rewarding for me because I feel like I can actually apply what I’ve learned from lectures like therapeutic knowledge to practice and it really like helped me consolidate similarities and differences between different meds. So it was very helpful.*
Student 2, focus group 7.

Students also appreciated the structure used in the counselling sessions in the unit and noted that this gave them a clear structure to follow when counselling in their workplace.


*I have used those questions going into a counselling session with a patient to try and get them to open up about what they already know. Which I have found useful.*
Student 6, focus group 6.

Finally, many also noted that their confidence to counsel patients had increased after completing this unit. This was true even of those students who were already working in pharmacies, one of whom noted that:


*I’ve been working in community pharmacy for a few years, but I was never comfortable in counselling drugs... until I took this unit of study.*
Student 1, focus group 1.

## 4. Discussion

To our knowledge, this is the first study to examine the combination of self-reflection and peer assessment on simulated patient counselling activities in pharmacy students; this was achieved by conducting a qualitative evaluation of third year pharmacy students’ perceptions of these techniques on their learning experience. When mapped against the domains of self-determination theory, we can see that learning style and feedback were covered by all three domains of the framework but benefits in real life practice was only covered in one domain of the framework (competence). Our results have three important implications for the use of these techniques in developing pharmacy students’ patient counselling skills.

First, the effectiveness of these techniques could be improved by more closely linking patient counselling activities to real life practice (e.g., by integrating them into students’ pharmacy placements and involvement in mystery shopping exercises). Previous studies reinforce the value of this approach. For example, Hyvarinen et al. [44], conducted a survey study to examine the use of a communication in disciplines pedagogy for teaching communication in pharmacy in authentic work situations. Qualitative and quantitative data were collected from 481 pharmacy students and mentors. They note that “authentic work situations offer excellent opportunities to draw the attention of the students to the notion of key disciplinary specialised communication tasks”. Meanwhile, in a focus group study with 263 UK undergraduate and postgraduate arts, professional, business, and technical students, Drew [45] found evidence that students are motivated, especially by courses having, relevance to the real world and work. Finally, several authors [46,47,48,49] have recently shown that practical training and feedback increase students’ motivation to study and to rehearse professional skills, and to become more deeply engaged in the discipline. Our students noted similar real-life benefits of this counselling unit to their practice (including increased confidence, medication knowledge and consolidation and counselling skills). Including activities such as counselling a patient on a certain number of the medications over the course of their placement, obtaining feedback on their counselling from their preceptor and completing a self-reflection on their counselling performance in the workplace as part of their placement portfolio are some ways that this could be achieved. Further, addition of a mystery shopping component [50]—where students assume the role of the consumer and receive advice or counselling for a specific ailment or medication and then act as a peer educator by providing feedback to pharmacy staff after the encounter—into student placements in the early years of their degree could reinforce to students the importance and benefits of counselling skills. Such an approach has also been shown [50] to have additional benefits for pharmacy students, including increased knowledge regarding the ailments covered in the intervention, improved performance in an objective structured clinical examination (OSCE) and improved self-perceived clinical competency score. In focus groups, participating students reported that they enjoyed this type of placement, gained valuable knowledge and communication skills to assist them in becoming a better pharmacist and improved professional identity and development of empathy.

Second, there appears to be a need to improve the quality of feedback that students provide to peers. Several students in this study suggested that they do not find peer feedback useful (particularly when it is provided by their friends) and that they often do not provide honest, comprehensive, or critical feedback to their peers. There is a growing literature on effective peer assessment, and this points to some key actions that educators can take to improve the quality of peer assessment. These include training [51,52,53] students to conduct peer assessment and review of and feedback on their peer assessment [54], experience in peer assessment [55], and the design of the peer assessment process (adequate timing, small group work [56] and the opportunity to revise work in response to peer assessment [53,57]^.^ Students’ academic achievement [58] and thinking style [59] are also important, as students with a high level of academic achievement and higher executive thinking style have been found to be more skilful in peer assessment. Matching students on academic ability and adapting the peer assessment process to meet their abilities has been suggested to overcome these factors [54].

Interestingly and in contrast to our results, the effect of reciprocity in biasing peer assessing—i.e., lack of fairness in assessing others due to personal relationships—has been suggested to be minimal. In a study [60] examining the quality of peer assessment of individual group process skills (contribution to discussion and contribution to group development) among 169 medical students working in groups of 9 to 11 students, reciprocity effects accounted for just 1% of the variance of peer assessment scores.

Importantly, this literature points to some simple changes that can be made to increase the effectiveness of peer assessment in this unit. Examples include training students in effective peer assessment, monitoring their assessments of their peers and providing further guidance and feedback on this where needed, increasing students experience with peer assessment throughout their degree (e.g., by incorporating more peer assessment activities in other units and earlier in their degree) and assigning students to groups for peer assessment based on academic ability rather than allowing them to assess their friends. Students will be assigned a de-identified student to anonymously assess when the course is next run in 2022.

Last, while many students noted that they found assessing their own video more useful than assessment by their peers, there is also scope to improve the quality of their self-reflection. There are two key issues here, both of which result in suboptimal self-reflection and feedback, but in different ways. Firstly, some students noted that they are unnecessarily strict when evaluating their own performance, which could provide too much feedback to be useful. Meanwhile, others noted that they found ongoing reflection on their own performance to be tedious. This suggests that they were not fully engaging in and therefore obtaining the full benefits of self-reflection, resulting in too little feedback. The literature in this area points to a range of strategies that can increase the quality of students’ reflections and assist them to master this skill, addressing both these issues. These include exposure to reflective assignments early in the program of study [61], repetitive exposure to reflective tasks throughout their degree [61], providing structure to reflective activities and providing opportunities for reflection in action [62], reflection training [63], the provision of external feedback on student performance to provide context to and inform self-reflection and assessment [64] and responding to both external and internal motivations [64] to encourage students to engage in self-reflection and develop reflective practice skills. Some of these approaches are already included in the unit of study (including providing structure for reflective activities and opportunities for reflection in action, as well as the provision of external) or the broader University of Sydney pharmacy curriculum (exposure to reflective assignments early in the degree and repetitive exposure to reflective tasks throughout their degree). However, the incorporation of reflection training into the pharmacy program and the use of both extrinsic and intrinsic motivation by teaching staff could encourage students to engage in this task more seriously and thoughtfully and help to increase the quality of self-reflection here.

Finally, the global COVID-19 pandemic has had significant impacts on the delivery of pharmacy education, and it is important to consider the feasibility of the intervention described here post-COVID. The use of video recordings can be considered COVID-safe and facilitate self-reflection, peer and tutor feedback. The unit was run fully online in 2021 due to COVID lockdowns in Sydney.

### Limitations

As with all qualitative research, the generalizability of our results is a potential issue. This is particularly important if trying to apply these results to students in other health disciplines and with other degree structures (e.g., where communication is more embedded in the degree or there is a greater integration of university teaching and workplace learning activities). It is also possible that our sampling strategy resulted in recruitment of a particular “type” of student—someone who cares enough about this issue to give up a considerable period of time to participate and/or who has a particular vision about their education and what activities they do or do not find valuable. Additionally, it is possible that our results do not represent the “true” beliefs and values of our students. For example, students might not be telling the truth and, even if they were not being overtly deceptive, the social desirability bias [65] may have led them to state what they think the interviewer or other students in the focus group wanted to hear, rather than what they truly believe and value. Indeed, it is even possible for people to deceive themselves about what they “truly” believe and value. However, the fact that we discovered a rich range of opinion with regard to the learning activities in this unit and that thematic saturation was reached (with all themes complete and well-described) suggests that these are not major issues. Finally, this study sought only the views of students. While this, in itself, does not diminish our findings, educational initiatives are best evaluated through a range of lenses [66]—including the student, teacher and other educators—and incorporating other views could lead to a richer evaluation here.

## 5. Conclusions

Students’ perceptions of self-reflection and self- and peer-assessment centred on learning style, feedback, and benefits in real-life practice, and mapped well to self-determination theory. While students felt that their needs around learning style and feedback were well met, additional focus on benefits in real-life practice will strengthen the impact of these learning activities. Units in the later stages of the degree should build on benefits in real-life practice of the activities described here.

## Figures and Tables

**Figure 1 pharmacy-11-00005-f001:**
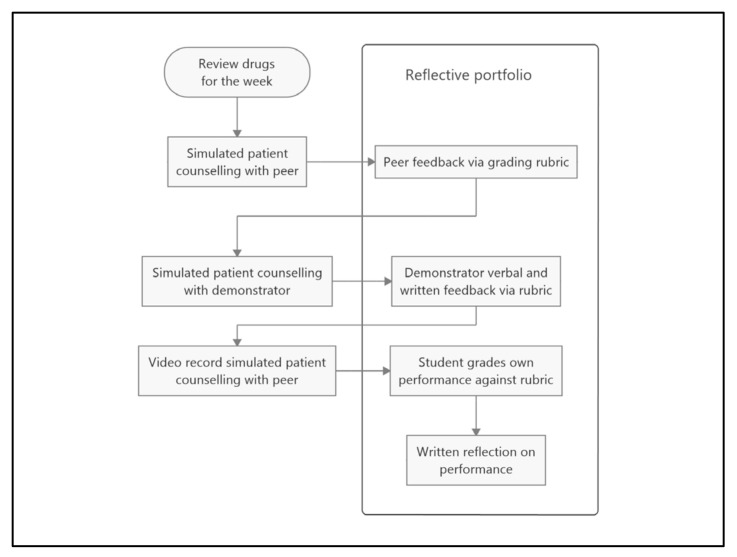
Structure of learning activity.

**Table 1 pharmacy-11-00005-t001:** Overview of themes as mapped against the components of self-determination theory.

		Self-Determination Theory		
		Autonomy	Competence	Relatedness
Themes	Learning Style	Gradual introduction to assessmentLearning through self-reflection videos	Learning through authentic assessmentIndividual learning compared to group learning	Learning through observation of best practice
	Feedback	Inconsistent feedback	Summative feedback	Perception of selfRelationship with peers informing peer assessment
	Benefits in real life practice		Benefits in real life practice	

## Data Availability

Not applicable.

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
