# Peer review of "Self-Reflection and Peer-Assessments Effect on Pharmacy Students’ Performance at Simulated Counselling Sessions"

_pharmacy, 2022, doi:10.3390/pharmacy11010005_

Round 1

Reviewer 1 Report

Thanks for your submission, it was an interesting read and can certainly add to the literature in this field. The manuscript is mainly well written, but there are some aspects to address (see below). The results are well presented with representative quotes throughout. 

Major comments to address:

1. Was there only one coder? If so, how did you ensure validation of the findings? (See Saldana 2013, page 35-36). More detail needs to be added to the “2.3 Data analysis” section than just “Detailed discussion amongst authors occurred throughout each stage of the analysis to determine final themes and subthemes”. Did this discussion involve blind coding and calculating kappa coefficients for example? If not, why not?

2. Regarding the themes, “Authentic assessment” as a subtheme of “Learning Style” is confusing. Should it be “Learning through authentic assessment”?

3. Tense use is confusing, make it consistent throughout. Examples include in the abstract, line 16, “These are…” should read “These were…”, and in the 2.1 Study setting section of the methods, change to past tense throughout (change “has” to “was”, “complete” to “completed”, “marks” to “marked”). 

4. Last line of the abstract: Make clearer regarding WHAT additional focus is required, presumably on verbal communication skills? 

Minor comments

1. Line 13: Change “…has impacted on their..” to “…has impacted their..”

2. Line 21: Change “…highlighted additional…” to “…highlighted that additional…”

3. Line 22: Insert gap between full stop and “Conclusion”

4. Line 29: First sentence of the introduction, it is proficiency in verbal communication that is the vital skill. To just state “verbal communication” could mean otherwise. 

5. Line 31: Fix gap between “the” and “International”

6. Lines 37-40: Possible cut-paste error here, please omit repeated words

7. Lines 71-81: Break this up, it is too long a sentence to read easily

8. Lines 90-91: The references cited are not recent. Either omit the term “recent” on line 89 or add some more recent references on self-assessment in pharmacy students, such as https://www.ajpe.org/content/86/4/8696

9. Line 406: Change “…student’s motivation…” to “…students’ motivation…”

10. Lines 414, 434, 466 and 467. Subscript the reference numbers

11. Line 478: Capitalise “covid”

Reviewer 2 Report

Thank you for the opportunity to review this manuscript. Given the importance of patient counseling as a pharmacist, this is a timely article. Plus, it was appreciated to see the juxtaposition of peer and self-reflection. Those are important skills to develop.

Here are some comments to consider:

Introduction

The introduction has a lot of great content, but it is extremely long and somewhat repetitive. I think some sentences are even repeated in full. Please consider having only 2-3 sentences cover the integration into professional standards and condensing or eliminating the remainder of the first 2-3 paragraphs. 

The simulation paragraph also has really good information, but it is quite long. Consider breaking this up into a few paragraphs and condensing. What are the key facts from the literature we need to know about peer- and self-evaluation, and how do these all relate to the subsequent gap analysis for your study?

Methods

The qualitative analysis is well-described. However, the manuscript would benefit from greater detail in several areas.

1) Details on how patient counseling and self-reflection is introduced prior to this course.

2) A figure overviewing the intervention assessed.

3) Details on whether the peer evaluations were de-identified or identified. It appears that they could pick whomever they wanted to review? Or, was it assigned? Making that clear would be very helpful. Were there any “screenings” of comments before release?

4) How did you develop the interview guide? What guided the generation of the guiding questions?

Those were the largest questions. The results are fine as well as the discussion. However, the manuscript overall could benefit from an author/co-author with an editing/condensing eye to tighten up the manuscript and make it more succinct and clearly written. There are some grammar issues as well - some word choices are not ideal. Again, an editing eye will likely fix this.

Author Response

Thankyou for your comments

Round 2

Reviewer 1 Report

Thanks for making revisions to your manuscript. However, more detail is still required in the methods section in terms of the process followed regarding generating the themes and subthemes. You should provide a temporal approach so that it is clear to the reader what process was followed, from initial data coding right through to theme generation. At present, it is not known what transpired in "discussion amongst the authors to refine codes". The first paragraph of section 2.3 needs to be re-written to make it clear from a step by step perspective how the themes were generated. This is especially important considering there was only one coder and therefore no opportunity to coder validation. 

Author Response

Thanks for your comments.

Methods section 2.3 has been rewritten to give a better outline of how the process of coding and refinement of themes took place and how team members were involved in this process